# Development of Slow-Releasing Tablets Combined with Persulfate and Ferrous Iron for In Situ Chemical Oxidation in Trichloroethylene-Contaminated Aquifers

**Geumhee Yun [1,†], Sunhwa Park [2,†], Young Kim [1] and Kyungjin Han [3,*]**

[1]  Department of Environmental Engineering, Korea University, Sejong 30019, Republic of Korea; rmal9292@korea.ac.kr (G.Y.); kimyo@korea.ac.kr (Y.K.)
[2]  National Institute of Environmental Research, Incheon 22689, Republic of Korea; parksu@korea.kr
[3]  Department of Environmental Engineering, Korea National University of Transportation, Chungju 27469, Republic of Korea
[*]  Correspondence: rudwls1009@ut.ac.kr; Tel.: +82-43-841-5356
[†]  These authors contributed equally to this work.

**Abstract:** Slow-releasing tablets combined with persulfate acting as an oxidant and ferrous iron acting as an activator were manufactured for in situ chemical oxidation. The trichloroethylene (TCE) removal efficiency according to the molar ratio of the oxidizer and activator in the 0, 0.5, 1, 1.5, 2, and 2.5 molar ratio (persulfate: ferrous iron) reactors were 15%, 89%, 90%, 82%, 71%, and 55%, respectively. In a batch reactor injected with an oxidation-activation combined tablet (OACT) and a liquid oxidizing/activator, the TCE removal efficiencies were 100% and 70%, respectively, showing that the tablet form had a high efficiency in contaminant removal. The evaluation of the dissolution characteristics and TCE removal efficiency of OACT 0.5 (tablet with a 1:0.5 molar ratio of persulfate to activator) and OACT 1.0 (tablet with a 1:1 molar ratio of persulfate to activator) under continuous flow conditions showed that the TCE removal efficiency of the OACT 1.0 column was approximately 1.4 times higher than that of OACT 0.5. The longevities of persulfate and ferrous iron of the OACT 1.0 tablet were 43.2 days and 41.7 days, respectively. Thus, OACT 1.0, which was manufactured effectively, was suitable for in situ slow-release chemical oxidation systems.

**Keywords:** chemical oxidation; slow-releasing tablet; oxidation-activation combined tablet; in situ slow-releasing chemical oxidation systems

## 1. Introduction

Conventionally, in situ groundwater chemical oxidation dissolves the oxidizing agent in an injection solution and supplies it to the groundwater in liquid form. Most in situ chemical oxidation processes target groundwater contaminated with petroleum and chlorinated aliphatic hydrocarbons (CAHs) by dissolving oxidizing agents, such as hydrogen peroxide and permanganate, in the liquid phase [1–5]. However, when a liquid-based oxidation process is applied in a contaminated aquifer, the concentration of contaminants may rebound. Hence, appropriate management strategies, such as periodic preparation and rejection of the oxidizing agent solution, are essential. Additionally, when applied for long-term operation, the removal efficiency and economic feasibility are reduced because of the tailing of the concentrated contaminants that may occur through back diffusion from the low-permeability layer in the latter half of the process application [6].

After injecting a liquid oxidizing agent into groundwater contaminated with CAHs, such as trichloroethylene (TCE), the contaminants have been reported to rebound after approximately 50 h, thus necessitating continuous injection [7]. Han et al. [8] applied a liquid-based management process to an aquifer contaminated with high nitrate concentrations, and a liquid was continuously injected in areas with rapid groundwater flow.

However, there are disadvantages, such as storing the liquid injection solution and the availability of spaces to occupy the site.

TCE, dichloroethylene, and vinyl chloride are often detected in the groundwater of large cities, such as Seoul, South Korea, which are characterized by high population densities. In this area, the risk of human exposure owing to contamination around subway stations with large floating populations is a concerning issue. Moreover, because the contaminated sites are located in a downtown area, the application of a large-scale remediation process is complex. Slow-release substances capable of controlled release allow the continuous supply of oxidizing agents for a long period with only one injection, minimizing the rebound and tailing of contaminant concentration and enabling efficient operation of the groundwater purification process [9]. Therefore, if an oxidation process using slow-release technology is applied, contaminated groundwater can be efficiently managed in urban areas. In a previous study, a candle-type, slow-release oxidizer combining permanganate and paraffin wax was developed and applied for groundwater remediation. However, this system exhibits several problems, such as insoluble paraffin by-products remaining in the well and the need for periodic removal of manganese dioxide scales from the candle surface [9–11].

Persulfate has the advantages of high solubility in water, high stability, and high oxidizing power. Moreover, it exhibits fewer problems with by-products generated during the reaction, which are disadvantages of permanganate, and can maintain a longer reaction time than hydrogen peroxide. In addition, the sulfate radical-based oxidation process is considered non-toxic, so it is an oxidation process that can effectively balance cost and environmental side effects compared to conventional hydroxyl radical-based oxidation processes [12–14]. Metals, such as ferrous iron, effectively promote this persulfate radical reaction [15,16]. Additionally, medicine-based, tablet-type, slow-releasing materials are environmentally friendly and produce no residual byproducts [6,17]. Therefore, in this study, we conducted an in situ groundwater remediation process using a tablet-based, slow-release oxidizer to manage hazardous contaminants in urban areas. To this end, we set up a tablet that combines persulfate, which can oxidize various organic substances, with ferrous iron as an activator to maximize radical reactions. The release and reduction characteristics of slow-release tablets were also evaluated.

## 2. Materials and Methods

### 2.1. Manufacture of Slow-Release Tablets

The slow-release tablets were based on microcrystalline cellulose (MCC) 101, hydroxypropyl methylcellulose 70 k (HPMC 70 k), and magnesium stearate, as reported by Han et al. [8] and were combined according to the ratios listed in Table 1. Sodium persulfate (Miwon Commercial Co., Ltd., Anyang, Republic of Korea) was used as an oxidizing agent for sustained-release tablets, and an aqueous ferrous iron [(FeSO$_4$·7H$_2$O($\pm$99%)] (Samchun, Seoul, Republic of Korea) was used as an activator. Specifically, the primary raw materials, persulfate or ferric iron, were mixed at approximately 60% of the total mass of the tablet. MCC 101 (BASF Chemical Co., Ltd., Ludwigshafen, Germany), HPMC 70 k (HPMC, Shin-Etsu Chemical Co., Ltd., Tokyo, Japan), and magnesium stearate (Namyoung Commercial Co., Ltd., Seoul, Republic of Korea) were mixed at 29%, 10%, and 1%, respectively, and then compressed into tablets using a Tablet Compression Machine (Good Price, Kimpo, Republic of Korea).

All materials were sieved with mesh #80 (nominal aperture 180 μm) and blended using a Turbular mixer (Turbular®, WAB, Muttenz, Switzerland) for 100 min. A hydraulic laboratory press (Carver® 3850, CARVER Inc., Wabash, IN, USA) tableted the mixture using a 10 mm rounded flat punch/die set under a tableting pressure of 2.0 tons. The physicochemical characterization of the manufactured tablets was evaluated through differential scanning calorimetry (DSC), X-ray diffraction (XRD), and Fourier transform infrared spectroscopy (FT-IR). The previous study described the detailed physicochemical characterization of the slow-releasing tablets [17].

**Table 1.** Component content in the prototype and combined tablets.

| Component | Prototype Slow-Releasing Oxidation Tablets [SROT(p)] | | Prototype Slow-Releasing Activator Tablets [SRAT(p)] | | Oxidation-Activation Combined Tablet 1:0.5 [OACT(0.5)] | | Oxidation-Activation Combined Tablet 1:1 [OACT(1.0)] | |
|---|---|---|---|---|---|---|---|---|
| | (mg/Tablet) | (Mass %) | (mg/Tablet) | (Mass %) | (mg/Tablet) | (Mass%) | (mg/Tablet) | (Mass%) |
| Sodium persulfate | 600 | 60 | N.I. [1] | N.A. [2] | 270 | 37.9 | 200 | 27.7 |
| Iron (II) sulfate heptahydrate | N.I. | N.A. | 600 | 60 | 158 | 22.1 | 233 | 32.3 |
| MCC 101 | 290 | 29 | 290 | 29 | 207 | 29.0 | 209 | 29.0 |
| HPMC 70 k | 100 | 10 | 100 | 10 | 71 | 10.0 | 72 | 10.0 |
| Magnesium stearate | 10 | 1 | 10 | 1 | 7 | 1.0 | 7 | 1.0 |
| Total (mg) | 1000 | 100 | 1000 | 100 | 713 | 100 | 721 | 100 |

Notes: [1] Not injected; [2] Not available.

A prototype slow-releasing oxidation tablet (SROT) and a slow-releasing activator tablet (SRAT) were manufactured according to the manufacturer's instructions. After evaluating SROT and SRAT, a tablet with a molar ratio of persulfate and activator of 1:0.5 [oxidation-activation combined tablet 1:0.5, OACT (0.5)] and a tablet with a molar ratio of persulfate and activator of 1:1 [oxidation-activation combined tablet 1:1, OACT (1.0)] were prepared.

*2.2. Soil Column Setup*

Three similar cylindrical soil columns (diameter 32 mm, length 80 mm, and total volume 64 cm$^3$) were constructed to evaluate the effects of OACT (0.5), OACT (1), and a control (OACT was not injected) on the chemical oxidation efficacy of TCE. Materials from an aquifer contaminated with CAHs, such as TCE, dichloroethylene, and vinyl chloride, were sieved through mesh #8 (nominal aperture 2.46 mm) to remove irregularly sized particles. Each column was packed with a sieved material to simulate the aquifer. The average bulk density and porosity were 1.55 g/mL and 0.31, respectively.

A continuously flowing cell was constructed with a serum bottle to mimic groundwater flow conditions in a well and to feed oxidants or activators released from SROA, SRAT, and OACT. An influent containing Br as a tracer was pumped from the reservoir into a continuously flowing cell at a 0.1 mL/min flow rate. The tablets were placed in a continuous-flow cell and the solution inside the cell was pumped into a soil column at a flow rate of 0.1 mL/min (Figure 1).

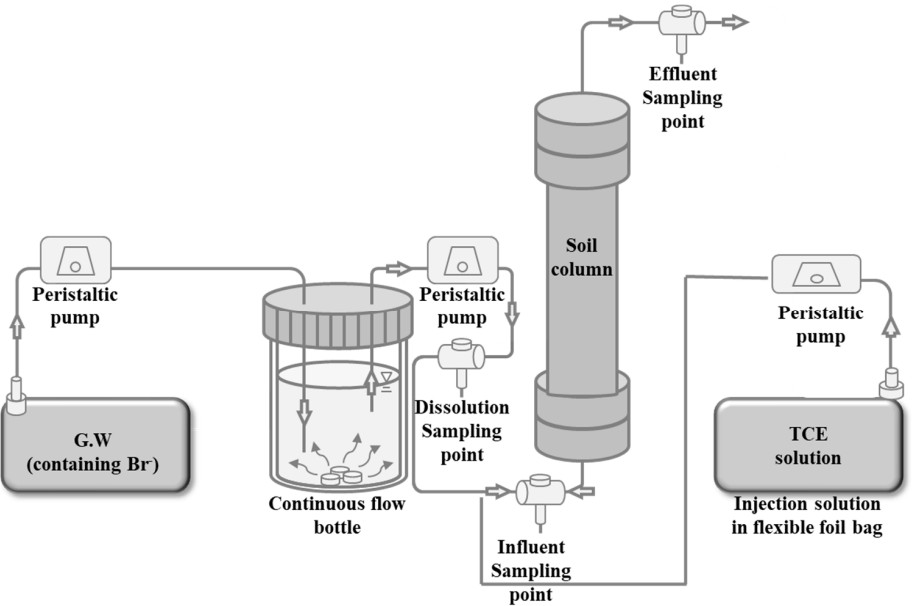

**Figure 1.** Schematic diagram of the column experiment.

### 2.3. Experiments Procedural

Experiments were conducted sequentially to evaluate the dissolution characteristics of slow-release tablets and the contaminant removal efficiency of the slow-release oxidation process under continuous-flow conditions simulating groundwater flow, as shown in Table 1. Using prototype tablets, a prototype tablet release test (PTRT) was performed to evaluate persulfate and ferrous release characteristics under continuous flow conditions. One tablet each of SROT and SRAT was injected into a continuous-flow vessel and distilled water was injected at a flow rate of 0.1 mL/mL. The prototype tablet batch test (PTBT) was performed to derive TCE removal efficacy by SROT and SRAT.

Theoretically, 1 mol of ferrous iron is required to activate 1 mol of persulfate radical reaction [18,19], as shown in Equation (1).

$$S_2O_8{}^{2-} + Fe^{2+} \rightarrow SO_4{}^{2-} \cdot + Fe^{3+} + SO_4{}^{2-} \tag{1}$$

Ferrous iron stimulates the activation of sulfate radicals during the oxidation reaction of persulfate. However, if present in excess, it can be used as a scavenger in radical consumption reactions [20,21]. Therefore, deriving an oxidizing agent and activator ratio is necessary to effectively promote radical reactions. Therefore, a combined tablet batch test (CTBT) was performed to optimize the mixing ratio of the oxidizer and activator in a batch reactor. In CTBT, we evaluated the oxidation characteristics of TCE under conditions of 0.5, 1, 1.5, 2, and 2.5 times the theoretical amount of ferrous iron required to activate persulfate. In this experiment, as shown in Equation (2), twice the amount of persulfate theoretically required to remove 10 mg/L of TCE was injected into all reactors. After deriving the optimal oxidizing agent to activator ratio, an OACT combining persulfate and ferrous iron was prepared using the manufacturing method described in Section 2.1.

$$3S_2O_8{}^{2-} + C_2HCl_3 + 4H_2O \rightarrow 6SO_4{}^{2-} + 2CO_2 + 9H^+ + 3Cl^- \tag{2}$$

A batch test was performed to evaluate the TCE removal efficiencies of SROT/SRAT, in which the oxidizing agent and activator were separated, and OACT, in which the two substances were mixed in one tablet. The experimental configurations are listed in Table 2. The amount of the tablet injection was designed such that each experimental group's total persulfate injection amount was the same.

**Table 2.** Composition of each experiment.

| | | Injected Tablet (Tablet) | | | | Flow Rate (mL/min) | Initial or Injected Solution Conc. (mg/L) | |
|---|---|---|---|---|---|---|---|---|
| | | SROT (P) | SRAT (P) | OACT (0.5) | OACT (1.0) | | Br- | TCE |
| Prototype tablet release test (PTRT) | SROT | 1 | N.I. [1] | N.I. | N.I. | 0.1 | N.I. | N.I. |
| | SRAT | N.I. | 1 | N.I. | N.I. | 0.1 | N.I. | N.I. |
| Prototype tablet batch test (PTBT) | Control | N.I. | N.I. | N.I. | N.I. | N.A. [2] | N.I. | 6.6 [3] |
| | SROT | 1 | N.I. | N.I. | N.I. | N.A. | N.I. | 6.5 |
| | SRAT | N.I. | 1 | N.I. | N.I. | N.A. | N.I. | 6.4 |
| | SROT + SRAT | 1 | 1 | N.I. | N.I. | N.A. | N.I. | 6.6 |
| Combined tablet batch test (CTBT) | Control | N.I. | N.I. | N.I. | N.I. | N.A. | N.I. | 6.7 |
| | SROT + SRAT | 2 | 2 | N.I. | N.I. | N.A. | N.I. | 7.8 |
| | OACT (0.5) | N.I. | N.I. | 4 | N.I. | N.A. | N.I. | 8.5 |
| | OACT (1.0) | N.I. | N.I. | N.I. | 6 | N.A. | N.I. | 8.9 |
| Combined tablet soil column test (CTST) | Control | N.I. | N.I. | N.I. | N.I. | 0.1 | $34 \pm 1.7$ [3] | $2.6 \pm 0.2$ |
| | OACT (0.5) | N.I. | N.I. | 3 | N.I. | 0.1 | $1075 \pm 65.5$ | $2.5 \pm 0.2$ |
| | OACT (1.0) | N.I. | N.I. | N.I. | 4 | 0.1 | $1120 \pm 72.8$ | $2.3 \pm 0.2$ |

Notes: [1] Not injected; [2] Not available; [3] Average value $\pm$ 95% confidence interval; TCE, trichloroethylene.

Finally, the combined tablet soil column test (CTST) was performed to evaluate the TCE removal efficacy of tablets containing combined persulfate and ferrous iron under continuous flow conditions. To equalize the amount of persulfate injected into the OACT (0.5) and OACT (1.0) columns, three OACT (0.5) tablets and four OACT (1.0) tablets were injected. The injection solution was designed to inject 10 mg/L of TCE, a contaminant, and 30 mg/L of $Br^-$ as a tracer.

### 2.4. Data Analysis

The dissolution rate in the batch tests was determined using the Higuchi model (Equation (3)):

$$M_t = K_H \times t^{1/2} \tag{3}$$

where $K_H$ ($T^{-1/2}$) is the Higuchi dissolution constant and $Mt$ (M) is the quantity of solute released at time t (T).

The data obtained were plotted as the cumulative percentage of persulfate or ferrous iron released against the square root of time [22]. This equation describes the solute released from pharmaceutical dosage forms, such as matrix tablets with water-soluble materials [23].

Several equations, such as power functions [24] and first-order decay [25], have been reported to fit the release profiles of slow-release candles to estimate the longevity of these systems. These studies used organic crystalline matrices and paraffin wax to prepare slow-release materials and proposed equations to express the dissolution and diffusion processes owing to matrix pore bonding. However, they differ from the HPMC and MCC used in the present study.

Yeum et al. [17] performed a fitting using nonlinear regression on an empirical saturation-type equation of a three-parameter logistic function (Equation (4)) to acquire the best-fitting plot of the mass ratio of the cumulative mass of the released solute versus the initial mass of solute ($M_r$) over the release time (T). $M_r$ versus t (T) plots were fitted using nonlinear regression with the SigmaPlot software (V. 12.0).

$$M_r = \frac{a}{1 + bt^c} \tag{4}$$

where $a$, $b$, and $c$ are the constants. All constants in this function are dimensionless.

$M_r$ was calculated by Equation (5):

$$M_r = \frac{Q\left(\frac{C_n + C_{n+1}}{2}\right)(t_{n+1} - t_n)}{M_i} \tag{5}$$

where $M_r$ is the mass ratio of the cumulative mass of persulfate or ferrous iron released versus the initial mass of persulfate or ferrous iron in SROT or SRAT, $Mi$ (M) is the initial mass of persulfate or ferrous iron in SROT or SRAT, Q (L3T-1) is the flow rate, $C_{n+1}$ (ML-3) is the concentration of persulfate or ferrous iron in the effluent from a continuous flowing cell at time $t_{n+1}$, and $C_n$ is the concentration of persulfate or ferrous iron in the effluent from a continuous flowing cell at time, $t_n$.

### 2.5. Analytical Methods

To measure the TCE concentrations, the samples were analyzed via headspace gas analysis using a Shimazu GC-17A (Kotyo, Japan) gas chromatograph (GC) equipped with a flame ionization detector. Further, 10 mL of the sample was removed from a 40 mL volatile organic acid vial, which was equilibrated in a shaking incubator at 20 °C. After equilibration, 200 μL of gas was sampled using a gastight syringe and injected into the GC. The specific operating conditions of the GC are described in Han et al. [8].

The $Br^-$ concentration was measured using an ion chromatograph (IC) equipped with a conductivity detector. Samples were prefiltered using a 0.35 μm syringe filter and placed in a Dionex AS-DV autosampler (Dionex Co., Sunnyvale, CA, USA). The IC system was operated at 1.2 mL/min, and the mobile phase was composed of a mixture of 3.5 mM

Na$_2$CO$_3$ and 1.0 mM NaHCO$_3$. The samples were delivered to the loop at 4.0 mL/min, with a flush factor of 10. A current of 24 mA was applied to the detector. The chromatographic separation was performed using a Dionex IonPac AS14 column (Dionex Co., Sunnyvale, CA, USA). Analytical-grade potassium bromide (99%), sodium nitrate (99%), and sodium nitrite (98%) (Samchun Chemical Co., Ltd., Seoul, Republic of Korea) were used for the column experiments and laboratory analyses.

Persulfate was analyzed using the absorbance method. Using a persulfate–iodine reaction, a coloring reagent was prepared by mixing 5 g/L sodium bicarbonate and 100 g/L potassium iodide. Further, 10 mL of the prepared coloring reagent was mixed with 200 μL of the sample, and the mixed solution was analyzed for ferrous iron at 400 nm using a DR6000 UV-VIS spectrophotometer (Hach Company, Loveland, CO, USA).

Specifically, ferrous iron was analyzed using the 255 program of the spectrophotometer with a 103769 analytical reagent.

## 3. Results and Discussion

### 3.1. Persulfate and Ferrous Releasing Characteristics in PTRT

Measuring the persulfate and ferrous iron release profiles of SROT and SRAT under continuous flow conditions is essential for designing a slow-release tablet system for the chemical oxidation of organic contaminants. The persulfate and ferrous iron release profiles were measured by monitoring the persulfate and ferrous iron concentrations in a continuously flowing cell. Two completely different phases of change in the temporal solute concentrations were observed during the dissolution test (Figure 2a). At the beginning of the SROT-flowing cell, the persulfate concentration rapidly increased, reaching a peak value at 45 h; subsequently, the concentration decreased rapidly. SRAT also showed a similar trend, with ferrous iron showing a maximum concentration after 55 h and then decreasing rapidly. This change in concentration resulted from the difference between the solute dissolution rate from the tablet and the influent dilution rate into the flowing cell. In controlled-release systems, an agent (i.e., an oxidized or reduced compound) is released rapidly, followed by a slower release rate [10,17,26]. Therefore, the rapidly decreasing persulfate concentration after 45 h may be due to reduced release rates. In other words, this suggests that the release rate of persulfate from the tablet is slower than the rate of water flowing into the flowing cell. During the first phase of rapid growth, the release rate may be greater than the dilution rate.

During the release period, the average persulfate and ferrous ion concentrations released from SROT and SRAT were approximately 3.7 ± 1.3 and 3.4 ± 0.8 mmol/L, respectively. The average oxidizing agent and activator concentrations of the tablet are critical for the slow-release oxidation process. As described in Equation (1), the theoretical molar ratio of the oxidizer to the activator was 1:1. Additionally, many studies have reported that pollutant removal efficiency differs by more than two times depending on the presence or absence of ferrous iron, an activator [15,16,21]. A similar trend was observed in the experimental results for PTRT. Thus, the concentrations of solutes released from SROT and SRAT indicated that the contaminants can be effectively managed by the sulfate radical reaction in the tablet-based in situ oxidation process. Additionally, the release rates of persulfate and ferrous iron could be effectively controlled in prototype tablets produced using this manufacturing method.

The PTBT analysis conducted to evaluate TCE removal efficiency using the prototype tablets showed that there was no change in the TCE concentration in the control reactor injected with only TCE and reactor-injected SOAT (Figure 2b). This indicated that none of the reactors showed TCE decomposition due to physical mechanisms, such as volatilization and ferrous iron, an accelerator. In the reactor injected with SROT, an oxidizing agent, approximately 68% of 10 mg/L TCE decomposed in 9 h, whereas in the reactor simultaneously injected with SROT and SRAT, TCE was decomposed completely in 5 d. This indicated that the activator ferrous iron acted as a radical accelerator in the persulfate oxidation reaction and effectively decomposed pollutants. CO$_2$ is a by-product of the

chemical oxidation of organic substances such as TCE [27,28]. Therefore, we measured $CO_2$ as a by-product to prove that the TCE decreased due to a chemical reaction. $CO_2$ was at average $0.12 \pm 0.04$ mg/L, $0.51 \pm 0.52$ mg/L, $0.27 \pm 0.12$ mg/L, and $3.68 \pm 4.78$ mg/L in the four PTBT reactors, respectively (Figure 2c). Therefore, the TCE removal rate and $CO_2$ generation tended to be proportional, which suggests that TCE was mineralized into $CO_2$ through the oxidation reaction of persulfate.

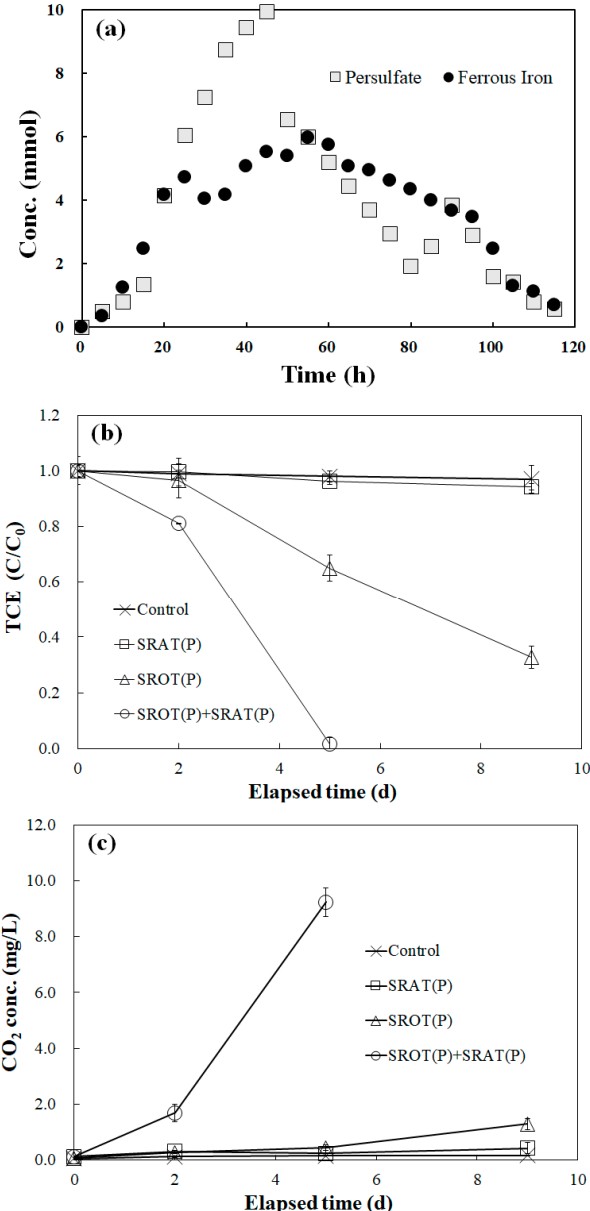

**Figure 2.** Release characteristics of prototypes SROT and SRAT under continuous flow conditions during the PTRT test (**a**) and TCE removal efficacy (**b**) and $CO_2$ concentrations (**c**) of prototype tablets in a batch reactor during the PTBT test.

## 3.2. Optimization Mixed Ratio of Oxidizer with Activator in CTBT

In the batch experiments conducted to evaluate the TCE removal efficiency using stoichiometric ratios of persulfate and the activator, the theoretical ferrous iron requirement according to Equation (2) was set to 0, 0.5, 1, 1.5, 2, and 2.5. After injecting the oxidizer and activator for 10 h, the TCE removal efficiencies at 0, 0.5, 1, 1.5, 2, and 2.5 molar ratio reactors were 15%, 89%, 90%, 82%, 71%, and 55%, respectively (Figure 3a). The low removal efficiency of TCE in the reactor without injecting ferrous iron as an activator resulted

from the direct chemical reaction of TCE with persulfate rather than a sulfate radical reaction [29,30]. Additionally, even when sufficient ferrous iron was injected, the TCE removal efficiency decreased, thus proving that sulfate radicals generated from persulfate through metal activity were consumed by excess ferrous iron. Therefore, these results suggested that the effective combination ratio of the oxidizing agent and activator would be 0.5–1 times the theoretical ratio of persulfate and ferrous iron. OACT (0.5) and OACT (1.0) tablets containing a combined oxidizing agent and activator were prepared according to the SROT and SRAT manufacturing methods.

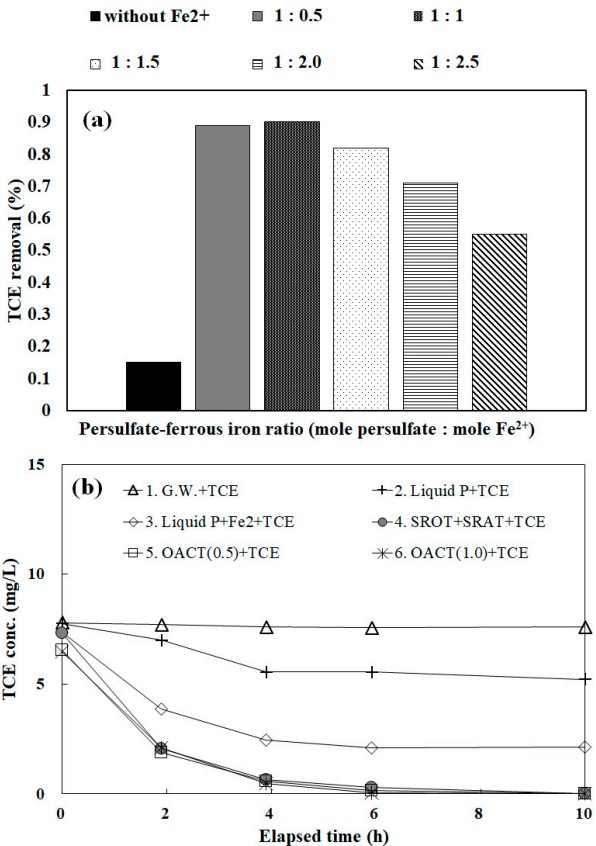

**Figure 3.** TCE degradation characteristics under various ferrous iron concentrations (**a**) and TCE degradation characteristics of prototype and combined tablets during the CTBT test (**b**).

A batch experiment was performed to evaluate the TCE removal efficiency of the OACT tablets. A reactor injected with a liquid oxidizing agent, oxidation tablet SROT, activator tablet SRAT, OACT (0.5), and OACT (1.0) were configured under the same conditions. The same mass of persulfate was injected into all reactors to accurately compare their efficiencies. In the control reactor, where only groundwater and TCE were present, the TCE concentration did not change for approximately 10 h, indicating no biochemical reactions of TCE due to dissolved substances present in groundwater (Figure 3b). In the reactor injected with only liquid persulfate, approximately 33% TCE was decomposed, whereas in the reactor injected with persulfate and ferrous iron simultaneously, the TCE removal efficiency was 71%. The injection of ferrous iron promoted the production of sulfate radicals in persulfate and was used to decompose pollutants. However, the oxidation reaction stopped after 6 h in both the liquid reactors.

This result presents a problem with the liquid injection method. Suppose that high oxidizing agent and activator concentrations are temporarily injected into the liquid. In such a case, they are not used efficiently to remove the target contaminants and can cause radical consumption and oxygen exchange rather than oxidation/promotion reactions. This indicates inefficient consumption of the activator through oxidation reactions with

oxygen. However, all three reactors into which the tablets were injected showed a similar trend, with 7 mg/L of TCE decomposed in approximately 6 h. This proved that the supply of oxidation/activators in the form of tablets in the in situ oxidation process to remove pollutants was more effective for decomposition than the conventional liquid injection method. Additionally, the results of injection did not differ through OACT, which is a combined oxidizer and activator, versus SROT with SRAT, suggesting that using combined tablets is efficient for the convenience of maintenance and production.

### 3.3. Evaluation of TCE Removal Efficacy in CTST

Soil column tests were performed to evaluate the oxidation efficacy of TCE using OACT (0.5) and OACT (1). The breakthrough curves of bromide (tracer) and TCE were obtained by measuring the effluent concentrations (Figure 4). For example, the same breakthrough curve C* ($C_{Eff}/C_{inj}$, effluent concentration divided by influent concentration) for Br and TCE indicated that TCE was not retarded (i.e., R = 1 for TCE). The lower C* of TCE compared with that of Br suggested that TCE may be adsorbed on the surface of the solid particles during transport through the column. The average C* of bromide in the three columns was 0.95 after approximately 2 d (two pore volumes), and 95% of the injected concentration was detected in the column effluent. The TCE_control column, in which only TCE was simultaneously injected, showed a C* value of 0.25, and only 25% of the injected concentration was detected at the outlet. Based on the calculation of the ratio of the 50% mass recovery time of TCE divided by the 50% mass recovery time of Br, the retardation factor of TCE was 1.49, indicating that TCE moves approximately 49% slower in the soil column via physical mechanisms, such as adsorption.

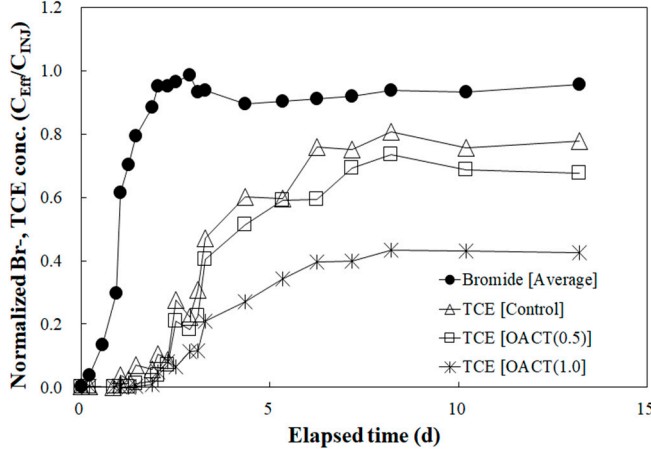

**Figure 4.** Normalized bromide and TCE concentrations in the soil column effluent during the CTST.

In the OACT (0.5) column, TCE showed a similar trend to the TCE_control column until approximately 4 d, but was detected to be approximately 0.1 times lower than that of the control column C*. C* in the OACT (1.0) column was significantly different from that in the other two columns after 3 d. The significantly lower C* value compared with that of the control column suggested that TCE degradation occurred through a chemical mechanism (oxidation) in addition to physical adsorption in the OACT (1.0) column. The mass recovery percentages of Br (average of 3 columns) and TCE in the control, OACT (0.5), and OACT (1.0) columns were calculated at 96% ± 2%, 61%, 50%, and 31%, respectively.

Significantly lower TCE C* values and mass recovery percentages of TCE compared to the tracer and control columns provided strong evidence of chemical oxidation induced by both the OACT (0.5) and OACT (1.0) columns. The lower removal efficiency of the column compared with that of the batch reactor suggested that the iron released from the OACT (0.5) tablet was likely consumed by oxygen or soil organic matter rather than by promoting sulfate radicals under the continuous injection of contaminated groundwater. However, in the case of the OACT (1.0) tablet, despite unnecessary consumption of ferrous iron, a

sufficient amount of activator was supplied for sulfate radical generation, and it would have shown a higher TCE removal efficiency compared to the OACT (0.5) column.

### 3.4. Estimating Longevity of OACT (1.0)

The longevity of persulfate and ferrous iron was confirmed using the results of OACT dissolution from the continuously flowing cell described in Section 3.3. In the flow cell, persulfate and ferrous iron showed maximum concentrations for approximately 2 d, similar to the release characteristics of the existing 3.1 prototype tablet, and the average concentrations at this time were $5.1 \pm 2.3$ mmol/L (Figure 5a) and $5.6 \pm 3.4$ mmol/L (Figure 5b), respectively. After 6 d, the concentrations of both solutes were maintained below 0.2 mmol/L, and the molar ratio of persulfate and ferrous iron was close to 1:1.

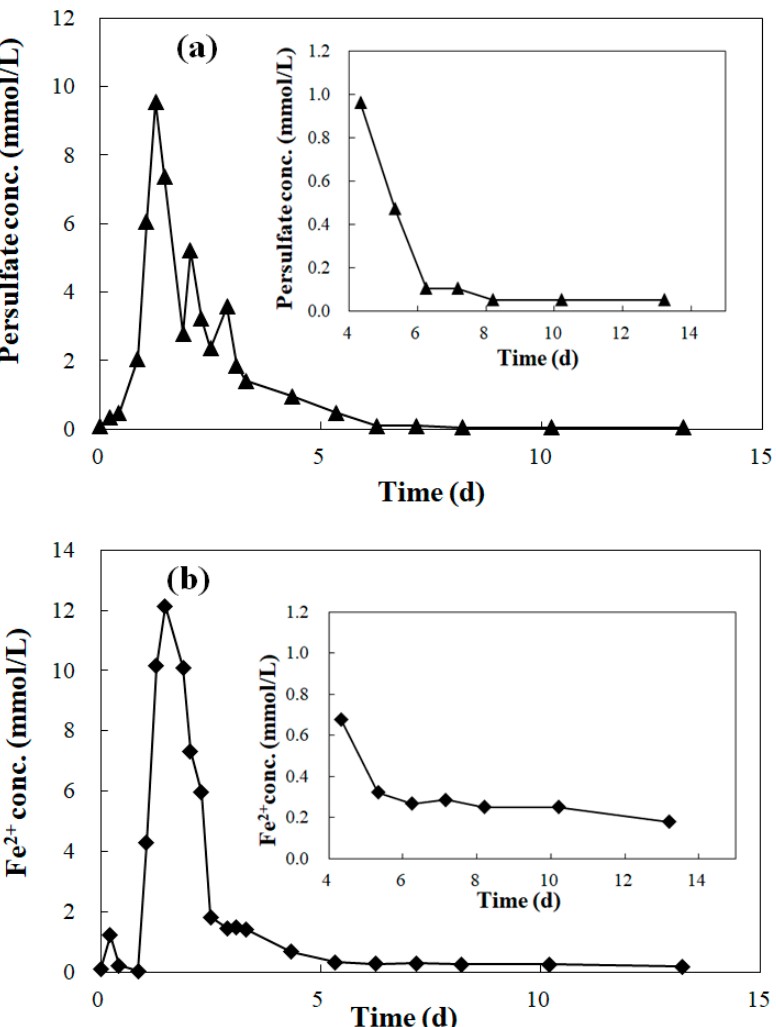

**Figure 5.** Release characteristics of the combined tablet of (**a**) OACT 1.0 PS and (**b**) ferrous iron under continuous flow during CTST.

For simplicity, previous studies defined the longevity of slow-release materials as the time taken to reach an Mr value of 0.80, as there was marginal change in Mr beyond 0.80. The predictions showed that OACT (1.0) persulfate and ferrous iron had 43.2 and 41.7 d of longevity, respectively (Figure 6). This study used small tablets (diameter 1.0 and height 0.6 cm), which have a shorter longevity than other large-scale release systems, and dissolution was found to last approximately 40 days. Considering that the longevity of slow-releasing material is generally dependent on size, the longevity is expected to increase if the tablet size is increased. The regression coefficient was >0.99 for all solutes.

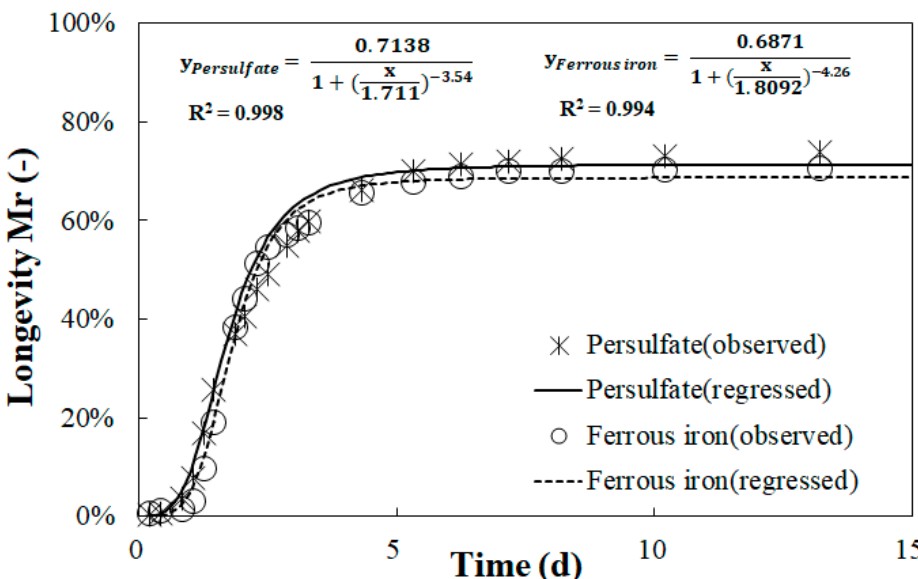

**Figure 6.** Observed and predicted $M_r$ values for persulfate and ferrous iron during continuous flow cell release experiments of CTST at a 0.1 mL/min flow rate.

## 4. Conclusions

This study aimed to develop a slow-release tablet that combined persulfate as an oxidizing agent and ferrous iron as an activator. Various batch and column experimental results showed that the molar ratio of the oxidation activator optimized for pollutant removal was 1:1. The dissolution characteristics of OACT (1.0) were evaluated, and persulfate and ferrous iron showed average concentrations of approximately $5.1 \pm 2.3$ and $5.6 \pm 3.4$ mmol/L, respectively, during the release period. This was close to the ideal molar ratio of the oxidizer to the activator of 1:1. The longevity of the manufactured OACT (1.0) indicated that the cumulative released mass of persulfate and divalent iron reached 80% after 43.2 and 41.7 d, respectively. Therefore, OACT, which can remove pollutants for several to tens of days with a single injection, would effectively manage contaminated groundwater that cannot be actively remediated, such as in urban areas or industrial complexes. In the future, optimization of the slow-releasing in situ chemical oxidation process is necessary through long-term monitoring of the release characteristics and pollutant removal efficacy of slow-releasing tablets at contaminated sites.

**Author Contributions:** G.Y.: conceptualization, writing—original draft preparation, formal analysis; S.P.: writing—reviewing and editing; Y.K.: investigation, writing—reviewing and editing; K.H.: investigation, writing—reviewing and editing, Supervision. All authors have read and agreed to the published version of the manuscript.

**Funding:** This research was supported by the Korea Ministry of Environment and the Technology Institute (KEITI) as "The Subsurface Environmental Management (SEM) project (2018002480005 and 2021002470002)". This research was also supported by a grant from the National Institute of Environment Research (NIER), funded by the Ministry of Environment (MOE) of the Republic of Korea (NIER-RP2022-01-01-069). This research was also supported by Basic Science Research Program through the National Research Foundation of Korea funded by the Ministry of Education (2022R1F1A1068350 and 2020R1I1A1A0106806). This article has not been reviewed by these agencies, and no official endorsement should be inferred.

**Data Availability Statement:** Data are contained within the article.

**Conflicts of Interest:** The funders had no role in the design of the study; in the collection, analyses, or interpretation of data; in the writing of the manuscript; or in the decision to publish the results.

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
