# Peer review of "Development of Slow-Releasing Tablets Combined with Persulfate and Ferrous Iron for In Situ Chemical Oxidation in Trichloroethylene-Contaminated Aquifers"

_water, doi:10.3390/w15234103_

Round 1
Reviewer 1 Report
Comments and Suggestions for Authors
This paper investigates development of slow-releasing tablets combined with persulfate and ferrous iron for in situ chemical oxidation in trichloro-3 ethylene-contaminated aquifers.
- Figure 1.B) How is possible that control has increment of concentration of TCE? The TCE cannot be over 1.0 (relative). Please measure in triplicates.
- Is there an explanation why concentration of persulfate from 10 mmol decreases to 6 in one measurement period?
- Why is there increment of persulfate concentration from 2 mmol to 4 mmol from 80 to 100 hours?
- In figure 2. there should be more data points from 0 to 6 hours. This is not good kinetic graph? You should make again these data points with less time interval.
- You are here not talking about mineralization of TCE and degradation byproducts. Can you make at least measurement of TOC to have data about mineralization?
- You have written models (full lines – predicted values). Where is data showing quality of model? Experimental vs Modeling data? Statistical analysis of model.
Comments on the Quality of English LanguageEnglish is OK.
Reviewer 2 Report
Comments and Suggestions for Authors
In this paper, the author studied the chemical oxidation of TCE in GW using persulfate and ferrous ions. Chemical oxidation of contaminants is a technology that has been successfully applied to the ex situ and in situ treatment of groundwater containing chlorinated organic solvents. This study contains very good results and needs to be published for a wider audience. I have a few but major suggestions for the improvement of the manuscript, which will be helpful for future studies.
· The author needs to add the advantages of applying persulfate and ferrous ions in the field, which must include their chemical stability, cost effectiveness, and the fact that their byproducts are less hazardous than those of other oxidants.
· Several oxidants, including hydrogen peroxide, Fenton's reagent (hydrogen peroxide combined with ferrous iron), permanganate and ozone, are currently reported for environmental remediation. Give a comparison table in the discussion section.
· Persulfate was used in some previous studies; how is yours different from other studies?
· Physicochemical characterization studies are missing.
· Kinetic study to determine the pathways or mechanism should be incorporated.
· Schematic diagram showing the layout of the column experiment is also missing.
Comments on the Quality of English LanguageModerate editing of English language required
Reviewer 3 Report
Comments and Suggestions for Authors
This manuscript has developed slow-releasing tablets combined with persulfate and ferrous iron for in situ chemical oxidation in trichloroethylene-contaminated aquifers. It is very interesting and innovative. In my opinion, it needs a minor revision. Here are some details.
1. The reaction process of tablets in the reaction system is related to the release rate, which is actually related to the characteristics of the drug, such as size, compactness during the production process, etc. However, the authors did not provide any relevant description or discussion in the paper. Please add if possible.
2. Please carefully check the citation of the references, for example: Han et al.(2022), I can not find it in the references list.
3. Please check Table 1 for horizontal parameters and units.
Round 2
Reviewer 1 Report
Comments and Suggestions for Authors
I do not have any more questions. The authors answeared to all my concerns.
Reviewer 2 Report
Comments and Suggestions for Authors
No further comments